# In Vitro Transcribed Artificial Primary MicroRNA for the Inhibition of Hepatitis B Virus Gene Expression in Cultured Cells [note 1]

**DOI:** 10.3390/microorganisms13030604

**Published:** 2025-03-05

**Authors:** Creanne Shrilall, Patrick Arbuthnot, Abdullah Ely

**Affiliations:** Wits/SAMRC Antiviral Gene Therapy Research Unit, Infectious Diseases and Oncology Research Institute (IDORI), Faculty of Health Sciences, University of the Witwatersrand, Johannesburg 2050, South Africa; creanne.shrilall@wits.ac.za (C.S.); patrick.arbuthnot@wits.ac.za (P.A.)

**Keywords:** Hepatitis B Virus, artificial primary microRNA, in vitro transcription

## Abstract

Available interventions for the management of chronic hepatitis B (hepB) exhibit limited efficacy and barriers to vaccination against the hepatitis B virus (HBV) have hampered prophylaxis programmes. Development of potent therapeutics capable of functional cure of chronic hepB thus remains a relevant medical objective. RNA interference (RNAi) can be exploited to effect potent and specific silencing of target genes through the introduction of RNA sequences that mimic the natural activators of the pathway. To achieve a therapeutic effect, artificial primary microRNAs (pri-miRNAs) have been used extensively to target various viruses, including HBV. To date artificial pri-miRNAs have exclusively been produced from DNA expression cassettes. Although this achieves impressive silencing, eventual translation of this platform to the clinic is complicated by the requirement for viral vectors to deliver DNA. Consequently, clinical translation has been slow. Recently, the use of in vitro transcribed RNA, specifically to produce mRNA vaccines at industrial scale, has gained significant interest. We therefore sought to evaluate the feasibility of using in vitro transcribed artificial pri-miRNAs for the inhibition of HBV gene expression. Artificial HBV-targeting pri-miR-31 sequences, which are highly effective when expressed in cells from a DNA template, demonstrated modest silencing of viral replication when incorporated into mRNA that was transcribed in vitro. Off-target effects were also observed. Characterisation revealed that intracellular processing of the artificial pri-miRNAs was inefficient and non-specific effects were caused by stimulation of the interferon response. Nevertheless, optimised nuclear delivery of the artificial pri-miRNAs should improve their processing and achieve better anti-hepB efficacy.

## 1. Introduction

Hepatitis B (hepB) is a highly transmissible liver disease that remains a great public health challenge [1]. The hepatitis B virus (HBV) infects hepatocytes and causes both acute and chronic hep B. In approximately 90% of perinatally infected infants and less than 5% of infected adults, the acute infection progresses to chronicity [2,3], reflecting limitations of an immature immune system for virus elimination. Chronic hepB is associated with severe complications such as cirrhosis and hepatocellular carcinoma [4], which contribute significantly to morbidity and mortality. It is estimated that, globally, 257 million people are chronic carriers of the virus and approximately 800,000 people die each year because of the persistent infection [1]. An effective vaccine against HBV is available which can induce a long-lasting protective antibody response against new infections in approximately 95% of healthy individuals [5]. However, overall vaccination coverage, especially in developing countries, remains low [6], and this represents a not insignificant barrier to eliminating the virus. Administration of the WHO-recommended birth-dose is particularly challenging in low- to middle-income countries [7]. The result is that the most vulnerable group of individuals remains at high risk for HBV infection and progression to chronicity. The current mainstream interventions for chronic HBV infection are the immunomodulator pegylated interferon alpha (PEG-IFN-α) and nucleoside/nucleotide analogue therapies. Although PEG-IFN-α may promote elimination of infected hepatocytes by promoting natural killer cell proliferation, activation, and antiviral activity [8], effectiveness is limited to a small subset of chronic carriers. It is estimated only 30% of individuals with chronic hepB will seroconvert and achieve reduced viral loads after PEG-IFN-α treatment [9,10]. Nucleoside/nucleotide analogue therapy functions by inhibiting the viral reverse transcriptase (RT), thereby reducing viral replication (reviewed in [11]). These drugs effectively reduce serum HBV DNA levels and may result in normalisation of liver inflammatory markers (reviewed in [12]). However, seroconversion of viral antigenemia is rarely achieved over the long-term [13], and this necessitates life-long use of the RT inhibitors. The need for effective therapy for chronic HBV infection therefore remains an important medical goal. Loss of HBV surface antigen (HBsAg) from the circulation of chronic carriers, defined as a functional cure, is considered the ideal endpoint of anti-HBV therapy [14]. HBsAg seroconversion and subsequent reactivation of liver inflammatory responses are thought to contribute to a better prognostic outcome by reducing long-term risks of chronic hepB [14,15]. Therapies aimed at reducing HBsAg therefore hold significant promise for treatment of chronic HBV infection. The RNA interference (RNAi) pathway has been used extensively to inhibit target gene expression, and the silencing mechanism can effect robust inhibition of replication viruses such as HBV [16,17,18,19,20,21,22,23,24,25,26,27]. We have previously exploited the RNAi pathway, through the use of artificial microRNAs (miRNAs), to silence HBV replication effectively [23,24].

miRNAs are the natural effectors of the RNAi pathway and their role as gene regulators is integral to normal cell functioning. miRNA genes are transcribed as single stranded primary miRNA (pri-miRNA) sequences that are cleaved by the microprocessor complex to form precursor miRNAs (pre-miRNAs), comprising stem loops approximately 70 nucleotides in length [28]. The pre-miRNAs are then transported to the cytoplasm where their loops are removed to form 20 bp miRNA duplexes with 2 nucleotide 3′-overhangs [29,30,31]. The duplexes are incorporated into the RNA induced silencing complex (RISC). One of the strands of the duplex is removed to leave the mature miRNA or guide strand in the complex. The mature miRNA guides RISC to complementary mRNA, which is degraded or translationally repressed, thereby silencing the gene encoded by the targeted mRNA. The RNAi pathway can be reprogrammed to silence genes of interest, by introducing mimics of pri-miRNAs, pre-miRNAs or miRNA duplexes into cells. Both synthetic [16,17,18,19,20] and expressed [21,22,23,24,25,26,27] RNAi activators have successfully been used to counter HBV replication. Synthetic RNAi activators generally belong to the class of chemically synthesised small interfering RNAs (siRNAs, 19–21 bp duplex with 2 nucleotide 3′-overhangs) and mimic the miRNA duplex. In contrast, expressed RNAi activators are generally designed to mimic pre-miRNAs (short hairpin RNAs and artificial pre-miRNAs) and pri-miRNAs (artificial pri-miRNAs). Because of their relative ease of manufacture and controlled delivery, synthetic siRNAs have progressed most rapidly with several anti-HBV candidates already in clinical trial [32,33,34,35,36,37,38]. However, the cost of producing chemically synthesised siRNAs is prohibitive in many settings and sequence fidelity cannot be guaranteed. Synthetic RNA produced by in vitro transcription potentially overcomes the limitations of chemical synthesis. To this end we have evaluated the production of in vitro transcribed artificial pri-miRNAs for the inhibition of HBV replication. Previously described artificial pri-miRNAs targeted to the *X* open reading frame of HBV (*HBx* ORF) [23,24], were adapted for production using in vitro transcription. Pri-miRNAs are typically expressed from RNA polymerase II promoters that regulate mRNA transcription. We thus aimed to evaluate the utility of in vitro transcribed artificial pri-miRNAs, which closely resemble endogenous pri-miRNA containing 5′ cap structures and polyA tails, as inhibitors of HBV.

## 2. Materials and Methods

### 2.1. mRNA Expression Vector Design

mRNA expression vectors were designed to serve as templates for in vitro transcription to generate monomeric or trimeric artificial pri-miR-31 sequences that target regions within the *HBx* ORF [23,24]. The *HBx* sequence is a convenient target to inhibit HBV replication as it is common to all the viral transcripts. The mRNA expression vector (pmRNA, TriLink Biotechnologies, San Diego, CA, USA) contained a T7 promoter, 5′ and 3′ UTR and a downstream sequence encoding a stretch of 52 adenosines. The vector was modified to include a CMV promoter and polyadenylation signal (pT7-CMV-pA) to allow in situ intracellular gene expression from the plasmid following transfection. *Nhe*I and *Sal*I restriction sites, amongst others, were introduced between the 5′ and 3′ UTR sequences to enable insertion of the artificial pri-miRNA sequences. Monomeric and trimeric pri-miR-31 encoding sequences were excised from the pCMV-pri-miR-31 plasmids [23,24] with *Nhe*I and *Sal*I and inserted into pT7-CMV-pA. Integrity of the mRNA expression vectors was confirmed by restriction mapping and sequencing (Inqaba biotec, Pretoria, South Africa).

### 2.2. In Vitro Transcription, Capping and mRNA Purification

The pT7-pri-miR-31/5, -31/8, -31/9, -31/589, and -31/895 plasmids were linearised with *Bpi*I and purified by ethanol precipitation. The linearised plasmid DNA was subsequently used for in vitro transcription of artificial pri-miR-31/5, -31/8, -31/9, -31/589, and -31/895 sequences using the TranscriptAid T7 High Yield Transcription Kit (Thermo Fisher Scientific, Waltham, MA, USA). In vitro transcribed RNA sequences were purified by LiCl precipitation. To remove dsRNA contaminants the prepared RNA was further purified using a cellulose-based method [39], and precipitated using LiCl. The purified RNA was subsequently capped using the Vaccinia Capping Enzyme System (New England Biolabs, Ipswich, MA, USA) according to the manufacturer’s instructions. Integrity and sizing of in vitro transcribed RNAs were assessed by formaldehyde agarose gel electrophoresis or using the 5200 Fragment Analyzer (Agilent, Santa Clara, CA, USA).

### 2.3. Cell Culture

Liver-derived HepG2-hNTCP cells were cultured in DMEM/F12 medium (Gibco™, Thermo Fisher Scientific, Waltham, MA, USA) containing 10% FBS, penicillin (200 U/mL) and streptomycin (200 μg/mL), G418 (400 μg/mL), insulin (5 μg/mL), hydrocortisone (50 μM), and HEPES (pH 7, 10 mM). Kidney-derived HEK293T cells were cultured in DMEM (high glucose) (Gibco™, Thermo Fisher Scientific, Waltham, MA, USA) containing 10% FBS, and penicillin (100 U/mL) and streptomycin (100 μg/mL). The cultured cells were maintained under standard conditions in a humidified incubator at 37 °C with 5% CO_2_.

### 2.4. Transfections

To assess silencing by artificial pri-miRNA sequences expressed from plasmid DNA, HepG2-hNTCP cells were co-transfected with 400 ng pT7-pri-miR-31 expression vectors and 50 ng of the HBV replication-competent plasmid pCH-9/3091 [40]. A plasmid expressing eGFP was also added to the transfection mix to assess transfection efficiency. Original pCMV-pri-miR-31 vectors were also used for comparison. Transfections were carried out in triplicate using Lipofectamine™ 3000 Transfection Reagent (Invitrogen™, Thermo Fisher Scientific, Waltham, MA, USA). Transfected cells were analysed by fluorescence microscopy two days post-transfection. Culture medium was harvested at that time and secreted HBsAg levels determined using the Monolisa™ HBs Ag ULTRA kit (Bio-Rad, Hercules, CA, USA).

Efficacy of in vitro transcribed artificial pri-miR-31 was assessed by transfection of cultured HepG2-hNTCP cells. Cells were grown on collagen coated plates which were prepared using Collagen I (rat tail) (Gibco™, Thermo Fisher Scientific, Waltham, MA, USA), according to the manufacturer’s instructions. Forty-eight hours after seeding cells, at 60–70% confluency, were co-transfected with 50 ng of a plasmid expressing the reporter mCherry (pAAV-minCMV-mCherry [41]) and 50 ng of pCH-9/3091. Alternatively, cells were transfected with the mCherry plasmid and a luciferase reporter of HBV expression (psiCHECK-HBx, [42]). Plasmid DNA transfections were carried out using Lipofectamine™ 3000 Transfection Reagent according to the manufacturer’s instructions (Invitrogen™, Thermo Fisher Scientific, Waltham, MA, USA). The culture medium was changed one day later, and the cells were transfected, in triplicate, with 500 ng of eGFP mRNA or in vitro transcribed artificial pri-miR-31 sequences using Lipofectamine™ MessengerMAX™ Transfection Reagent (Invitrogen™, Thermo Fisher Scientific, Waltham, MA, USA). Inhibition of HBV gene expression was assessed 24- and 48-h post transfection. Culture medium was harvested from cells transfected with pCH-9/3091 and secreted HBsAg levels measured using the Monolisa™ HBs Ag ULTRA kit (Bio-Rad, Hercules, CA, USA). Cells transfected with psiCHECK-HBx were harvested and analysed for luciferase expression using the Dual-Luciferase^®^ Reporter Assay System (Promega, Madison, WI, USA).

### 2.5. Northern Blot Analysis

To determine whether in vitro transcribed artificial pri-miRNAs are processed into the expected mature miRNA guide sequences, the pri-miR-31/5 sequence was transfected into HEK293T cells, and the total RNA was assessed by Northern blot hybridisation analysis. HEK293T cells were transfected with 10 µg of in vitro transcribed eGFP mRNA, in vitro transcribed pri-miR-31/5 or pCMV-pri-miR-31/5. The RNA transfections were carried out using Lipofectamine™ MessengerMAX™ Transfection Reagent whereas the DNA transfection was performed using the Lipofectamine™ 3000 Transfection Reagent. The eGFP or pri-miR-31/5 transfected cells were analysed by fluorescence microscopy one day after transfection and the total RNA was isolated using TRIzol™ Reagent. The pCMV-pri-miR-31/5 transfected cells were analysed two days after transfection followed by total RNA extraction. For Northern blot hybridisation analysis, a radioactively labelled molecular weight ladder and probe were generated. To generate the molecular weight ladder, several oligonucleotides (7 nt, 10 nt, 11 nt, 21 nt, 22 nt, 38 nt) were radioactively labelled separately and purified in a single Sephadex column. To generate the radioactively labelled probe for the detection of the miR-31/5 guide sequence, a complementary oligonucleotide was labelled and purified. The molecular weight ladder and 40 µg of total RNA were heat denatured and separated on a 20% denaturing polyacrylamide gel (8 M urea, 10× TBE) at 300 V. The gel was stained with ethidium bromide and imaged using a gel documentation system. Thereafter, the separated samples were electroblotted onto a nylon membrane and fixed to the membrane by UV crosslinking (Ultraviolet Crosslinker system, UVP, Upland, CA, USA). The region containing the ladder was cut from the rest of the membrane and exposed to a phosphorimager plate, which was imaged one day later (FLA-7000 imaging system, Fujifilm, Tokyo, Japan). The membrane containing the RNA samples was pre-hybridised with the PerfectHyb Plus buffer (Sigma-Aldrich, St. Louis, MO, USA) at 45 °C for 5 min with rotation. Hybridisation with the radioactively labelled probe was carried out overnight at 45 °C. Following overnight hybridisation, the membrane was subjected to two low stringency washes (2× SSC, 0.1% SDS at 25 °C for 5 min) and two high stringency washes (0.5× SSC, 0.1% SDS at 42 °C for 20 min). The membrane was exposed to a phosphorimager plate and imaged one week later using the FLA-7000 imaging system.

### 2.6. Assessment of Cellular Toxicity

In vitro transcribed artificial pri-miR-31 sequences were used to transfect HEK293T cells, and cytotoxicity was assessed by performing a 3-(4,5-dimethylthiazol-2-yl)-2,5-diphenyltetrazolium (MTT) assay. Briefly, HEK293T cells were seeded in a 96-well plate at a confluency of 40%. Forty-eight hours later, the cells were transfected, in triplicate, with 100 ng of in vitro transcribed eGFP mRNA or pri-miR-31 sequences using Lipofectamine™ MessengerMAX™ Transfection Reagent (Thermo Fisher Scientific, Waltham, MA, USA). Untransfected cells, mock transfected cells and cells treated with Triton X-100 were included as controls. Twenty-four hours post-transfection 20 µL of 5 mg/mL of MTT in 1 × PBS was added to the cells and incubated at 37 °C for 1 h. The supernatant was removed and the crystallised formazan dissolved in DMSO. Absorbance was then measured at a ratio of 570 nm to 655 nm using the iMARK™ microplate absorbance reader (Bio-Rad, Hercules, CA, USA).

### 2.7. Assessment of Interferon Response Stimulation by Dual Luciferase Assay and RT-qPCR

Stimulation of the interferon response in HEK293T cells transfected with in vitro transcribed artificial pri-miR-31 sequences was assessed by dual-luciferase assay and RT-qPCR. HEK293T cells were seeded in 24- or 48-well plates at a confluency of 45% and transfected one day later. For RT-qPCR analysis HEK293T cells in 24-well plates were transfected with 500 ng of poly I:C, in vitro transcribed eGFP mRNA or in vitro transcribed pri-miR-31 sequences. The transfections were carried out in triplicate using the Lipofectamine™ MessengerMAX™ Transfection Reagent. Twenty-four hours post-transfection the culture supernatant was discarded and total RNA extracted from the cells using Trizol^®^ Reagent (Invitrogen™, Thermo Fisher Scientific, Waltham, MA, USA). *IFN-β*, *OAS1*, *MxA* and *GAPDH* gene expression was measured using the Luna^®^ Universal One-Step RT-qPCR Kit (New England Biolabs, Ipswich, MA, USA) with the CFX96 Touch™ Real-Time PCR Detection System (Bio-Rad, Hercules, CA, USA). Primers used for RT-qPCR have been described before [42], with the following cycling conditions: reverse transcription at 55 °C for 10 min, initial denaturation at 95 °C for 1 min, 44 cycles of denaturation at 95 °C for 10 s, extension at 60 °C for 30 s. Melting curve analysis was carried by measuring amplicons denaturation across 65 °C to 95 °C, at increments of 0.5 °C for 5 s (plate read at each temperature increment).

For the dual luciferase assay HEK293T cells in 48-well plates were co-transfected with 50 ng of pUCAsc-ISRE×3, 50 ng of phRL-CMV (Promega, Madison, WI, USA) and 100 ng of poly I:C, in vitro transcribed eGFP mRNA or in vitro transcribed pri-miR-31 sequences. The transfections were carried out in triplicate using Lipofectamine™ 3000 Transfection Reagent. Forty-eight hours after transfection cells were harvested and luminescence measured using the Dual-Luciferase^®^ Reporter Assay System (Promega, Madison, WI, USA).

## 3. Results

### 3.1. Design of Artificial pri-miR-31 Sequences for In Vitro Transcription

Previously described artificial pri-miR-31 sequences with guide strands targeted to the *HBx* ORF [23,24], were adapted to be expressed from a modified pmRNA expression vector (Figure 1A). The pmRNA vector contains a T7 promoter to drive in vitro transcription using T7 RNA polymerase. It also contains a synthetic 5′ UTR and 3′ UTR derived from the murine haemoglobin alpha chain mRNA. A stretch encoding 52 adenine residues was included to create a polyA tail during in vitro transcription. The vector also contains a CMV promoter and pA signal that allows inserted transgenes to be expressed from the plasmid DNA itself. The monocistronic pri-miR-31/5, -31/8 and -31/9, and the polycistronic pri-miR-31/589 and -31/859 sequences were inserted into the pmRNA vector to allow their synthesis using in vitro transcription. Because the artificial pri-miR-31 sequences are not translated, the final vectors lacked a Kozak sequence.

### 3.2. Inhibition of HBV Gene Expression by the Artificial pri-miR-31 DNA Expression Vectors

To assess whether the artificial pri-miR-31 sequences placed in the context of an expression vector for in vitro transcription were still functional, the plasmid DNA was used to transfect the liver-derived HepG2-hNTCP cell line [43]. As a comparator, the original artificial pri-miR-31 expression systems, which effectively silence HBV gene expression [23,24], were also evaluated. HBsAg is a convenient marker to assess HBV gene expression, because the antigen is produced in high concentrations during replication and can conveniently be measured. The HBV replication-competent plasmid pCH-9/3091 was used as the target and co-transfected with the pCMV-pri-miR-31 or pT7-pri-miR-31 vectors. pCI-neo (Promega, Madison, WI, USA) and the empty pmRNA vector were used for mock transfections as these plasmids comprise the backbones of the pCMV-pri-miR-31 and pT7-pri-miR-31 vectors. Equivalent and good transfection efficiencies were confirmed by detecting eGFP expression from the co-transfected reporter plasmid. HBsAg secretion into culture supernatant was measured, and demonstrated that the pT7-pri-miR-31 vectors caused robust silencing of HBV gene expression (Figure 1B, pT7-pri-miR-31/5 *p* = 0.0003; pT7-pri-miR-31/8 *p* = 0.0005; pT7-pri-miR-31/9 *p* = 0.0098; pT7-pri-miR-31/589 *p* = 0.0005; pT7-pri-miR-31/895 *p* = 0.0004; d.f. = 4). However, overall silencing was not as efficient as that achieved by the original pCMV-pri-miR-31 vectors. This is not unexpected as the pmRNA backbone vector is not optimised for expression in mammalian cells. Nevertheless, the data indicate that the artificial pri-miR-31 sequences within the context of the mRNA expression vector are functional, and the production of in vitro transcribed pri-miR-31 mimics would yield functional RNAi activators.

### 3.3. Inhibition of HBV Gene Expression by Artificial pri-miR-31 Sequences

Following linearisation of pT7-pri-miR-31 vectors with the type IIs restriction enzyme *Bpi*I, the templates were used in standard in vitro transcription reactions. An mRNA sequence that codes for eGFP was also produced. The RNAs were purified using a cellulose-based method to eliminate dsRNA [39], which may cause unwanted stimulation of the innate immune response. The RNAs were subsequently capped enzymatically using the Vaccinia Capping Enzyme to produce RNA sequences with 5′ caps and 3′ polyA tails that resemble endogenous mRNAs and pri-miRNAs produced by RNA polymerase II. The in vitro transcribed RNAs were intact and of good quality after multiple rounds of purification (Appendix A).

The silencing efficiency was assessed by transfecting HepG2-hNTCP cells sequentially with pCH-9/3091 and then in vitro transcribed artificial pri-miR-31 RNA sequences. The transfection efficiencies were confirmed by microscopy (mCherry for plasmid transfection and GFP for RNA transfection) and the amount of HBsAg secreted was measured in the culture supernatants. Treatment of cells receiving the HBV target plasmid, pCH-9/3091, with the in vitro transcribed artificial pri-miR-31 sequences resulted in significant reductions in secreted HBsAg from HepG2-hNTCP cells (Figure 2A, Pri-miR-31/5 *p* = 0.0128; Pri-miR-31/8 *p* = 0.0084; Pri-miR-31/9 *p* = 0.0219; Pri-miR-31/589 *p* = 0.0113; Pri-miR-31/895 *p* = 0.0289 at 24-h and Pri-miR-31/5 *p* = 0.0012; Pri-miR-31/8 *p* = 0.0004; Pri-miR-31/9 *p* = 0.0027; Pri-miR-31/589 *p* = 0.0001; Pri-miR-31/895 *p* = 0.0014 at 48-h; d.f. = 4). Furthermore, the effect was enhanced 48 h post-transfection (12.7–30.3%) as compared to 24 h post-transfection (5.6–22.1%). However, the control in vitro transcribed eGFP mRNA also caused significant reduction in secreted HBsAg (Figure 2A, eGFP mRNA *p* = 0.0172 at 24-h and *p* = 0.0017 at 48-h; d.f. = 4). These findings indicate that the observed reduction in HBsAg levels is a result of non-specific inhibition, rather than sequence-specific silencing by the RNAi pathway. The reductions in HBsAg levels by the artificial pri-miR-31 sequences were greater than the reduction caused by the eGFP mRNA, suggesting that there is some sequence-specific silencing. When a commercially available GFP mRNA, produced with modified nucleotides and purified by HPLC (CleanCap^®^ EGFP mRNA, TriLink Biotechnologies, San Diego, CA, USA), was used there was minimal reduction in HBsAg levels (Appendix A). This result suggests that the in vitro transcribed artificial pri-miR-31 sequences can be further optimised to limit non-specific effects.

To delineate the difference between specific and non-specific inhibition of viral gene expression, a dual-luciferase reporter plasmid, psiCHECK-HBx, was used as the target of artificial pri-miR-31 silencing. This reporter plasmid has the *HBx* ORF inserted downstream of a Renilla *luciferase* sequence and as a result *Renilla* luciferase expression is indicative of targeted silencing of the *HBx* sequence. Firefly luciferase is also expressed from the plasmid but is independent of the *HBx* sequence. The ratio of the *Renilla* to Firefly luciferase activity is therefore a more accurate indication of silencing because Firefly luciferase activity adjusts for any unrelated variation. HepG2-hNTCP cells were transfected with psiCHECK-HBx followed by transfection of the in vitro transcribed artificial pri-miR-31 sequences. Transfection was confirmed by fluorescence microscopy where red fluorescence indicates plasmid transfection efficiency and green fluorescence is indicative of RNA transfection efficiency. *Renilla* and Firefly luciferase activities were measured in transfected cells and the ratios of the two plotted (Figure 2B). Based on relative luciferase activities, in vitro transcribed eGFP mRNA showed no statistically significant reduction. These data are more in line with the expectation that eGFP mRNA should not exert any effect on the *HBx* sequence and verifies the ability of this assay to distinguish specific silencing from non-specific effects. However, apart from pri-miR-31/5 (*p* = 0.0049 at 24-h and *p* = 0.0076 at 48-h; d.f. = 4) and pri-miR-31/8 (*p* = 0.0172 at 48-h; d.f. = 4), none of the other in vitro transcribed artificial pri-miR-31 sequences exhibited specific silencing of the *HBx* sequence. The inability of these transcripts to effect any sequence-specific silencing is unexpected but again indicates the need for improving the design of the system.

### 3.4. Processing Efficiency of the In Vitro Transcribed pri-miR-31/5

The artificial pri-miR-31/5 sequence and eGFP mRNA were in vitro transcribed, purified by cellulose chromatography and capped using the Vaccinia Capping Enzyme System (Appendix A). HEK293T cells were transfected with the in vitro transcribed RNAs and 24 h later, total RNA was extracted from the cells and analysed by Northern blot hybridisation. Cells were also transfected with plasmid DNA expressing artificial pri-miR-31/5 to serve as a comparator. The design of the artificial pri-miR-31/5 sequence is such that processing of this transcript by the miRNA biogenesis pathway should yield a guide sequence complementary to co-ordinates 1575–1595 of the HBV genome [22,23,24]. It is expected that the in vitro transcribed artificial pri-miR-31/5 sequence is recognised by the microprocessor complex (DGCR8 and Drosha) and processed to produce the artificial pre-miR-31/5, which should subsequently be processed by Dicer and incorporated into RISC. The final guide strand should then direct the complex to HBV transcripts, which are cleaved by RISC then degraded. Northern blot hybridisation with a probe complementary to the expected miR-31/5 guide sequence revealed a band corresponding to the putative mature miR-31/5 guide sequence (Figure 3). However, the miR-31/5 guide strand produced from the in vitro transcribed RNA was at much lower quantities than that produced from plasmid DNA. Northern blot analysis further revealed that much of the in vitro transcribed pri-miR-31/5 sequences remained unprocessed. This is in stark contrast to the pri-miR-31/5 sequences expressed from plasmid DNA, which all exist either as processed pre-miR-31/5 sequences or as the mature miR-31/5 guide. Together, these data indicate that in vitro transcribed artificial pri-miR-31 sequences are not efficiently processed in transfected cells. The first step of miRNA biogenesis is processing of pri-miRNA to pre-miRNA by the microprocessor complex in the nucleus. A likely explanation for the observed poor processing efficiency is that only a small fraction of the in vitro transcribed pri-miR-31 sequences transfected into the cells reached the nucleus. The reduced processing efficiency of the artificial pri-miR-31/5 sequence likely explains the low inhibitory capacity demonstrated by the in vitro transcribed sequences (Figure 2A,B).

### 3.5. Assessing Non-Specific Silencing of In Vitro Transcribed Artificial pri-miR-31 Sequences

The cytotoxicity of the artificial pri-miR-31 sequences was assessed by performing an MTT assay to measure the cultured cells’ viability after treatment. Artificial pri-miR-31 sequences and 5′ capped eGFP mRNA were in vitro transcribed, purified using cellulose-based chromatography, and used to transfect HEK293T cells. Transfection efficiency was confirmed by fluorescence microscopy and the percentage of viable cells assessed. Cells treated with Triton X-100 were used as a positive control for cytotoxicity and untransfected and mock transfected cells served as negative controls. Although transfection of eGFP mRNA reduced HEK293T cell viability significantly (*p* = 0.0492; d.f. = 4) (Figure 4A), at 2% the reduction was minor, and indicates that the in vitro transcribed RNAs did not induce gross toxicity in cultured cells (Pri-miR-31/5 *p* = 0.5177; Pri-miR-31/8 *p* = 0.0736; Pri-miR-31/9 *p* = 0.4415; Pri-miR-31/589 *p* = 0.0677; Pri-miR-31/895 *p* = 0.2367; d.f. = 4). The in vitro transcribed artificial pri-miR-31 sequences, similarly, did not result in significant cell death. The simplicity of the MTT assay has resulted in its widespread use, however the assay lacks the sensitivity to detect minor perturbations in normal cellular function. Therefore, although the MTT assay has demonstrated that the artificial pri-miR-31 sequences do not cause significant toxicity, it cannot detect the minute changes that may explain the observed non-specific silencing (Figure 2A,B).

Non-specific silencing may result from sequence-dependent and -independent stimulation of the innate immune response. More specifically, recognition of in vitro transcribed RNA by pattern recognition receptors may lead to stimulation of the interferon response and expression of interferon stimulated genes. Activation of the interferon response may cause non-specific inhibition of gene expression and could explain the observed non-specific effects of in vitro transcribed pri-miR-31 sequences. The capacity of the artificial pri-miR-31 sequences to stimulate the interferon response was first assessed using a luciferase reporter assay. The reporter plasmid, pUCAsc-ISRE×3, expresses Firefly luciferase under the control of 3 interferon-sensitive response elements (ISREs) which are activated by the interferon response factor class of transcription factors [44]. The plasmid, phRL-CMV (Promega, Madison, WI, USA), constitutively expresses *Renilla* luciferase and is independent of interferon stimulation. To assess activation of the interferon response, cellulose-purified artificial pri-miR-31 sequences were co-transfected with pUCAsc-ISRE×3 and phRL-CMV. The ratio of Firefly luciferase to *Renilla* luciferase activity in lysates of transfected cells were used as a measurement of interferon induction. Poly I:C is a positive inducer of the innate immune response and caused significant upregulation of Firefly luciferase activity when used to transfect HEK293T cells (Figure 4B). Significant increases in Firefly luciferase activity were also observed in cells that were transfected with artificial pri-miR-31/8 (*p* = 0.0225; d.f. = 4) and pri-miR-31/9 (*p* = 0.0366; d.f. = 4), indicating that these sequences stimulated the interferon response. Although Firefly luciferase activity was increased in cells transfected with artificial pri-miR-31/5, pri-miR31/589 and pri-miR-31/895, this difference was not statistically significantly different from untreated cells (*p* = 0.1623; 0.3851 and 0.0942, respectively; d.f. = 4). Nevertheless, the data support the idea that the in vitro transcribed RNA sequences stimulate the interferon response.

To further characterise interferon response stimulation by the in vitro transcribed artificial pri-miR-31 sequences, sensitive RT-qPCR was carried out to measure changes in *IFN-β*, *OAS1* and *MxA* gene expression. HEK293T cells were transfected with in vitro transcribed 5′ capped eGFP mRNA and artificial pri-miR-31 sequences. Using the more sensitive RT-qPCR assays revealed significant increases in *IFN-β* (Pri-miR-31/5 *p* = 0.0096; Pri-miR-31/8 *p* = 0.0075; Pri-miR-31/9 *p* = 0.0174; Pri-miR-31/589 *p* = 0.0033; Pri-miR-31/895 *p* = 0.0054; d.f. = 4), *OAS1* (Pri-miR-31/5 *p* = 0.0041; Pri-miR-31/8 *p* = 0.0138; Pri-miR-31/9 *p* = 0.0041; Pri-miR-31/589 *p* = 0.0140; Pri-miR-31/895 *p* = 0.0136; d.f. = 4) and *MxA* (Pri-miR-31/5 *p* = 0.0072; Pri-miR-31/8 *p* = 0.0138; Pri-miR-31/9 *p* = 0.0025; Pri-miR-31/589 *p* = 0.0051; Pri-miR-31/895 *p* = 0.0160; d.f. = 4) gene expression in cells treated with artificial pri-miR-31 sequences (Figure 4C–E). In the case of pri-miR-31/5 and pri-miR-31/8, the degree of stimulation was comparable to that of the positive inducer of the interferon response, poly I:C. Cells transfected with in vitro transcribed eGFP mRNA also exhibited increased expression of *OAS1* and *MxA*. A commercial GFP reporter mRNA (CleanCap^®^ EGFP mRNA, TriLink Biotechnologies, San Diego, CA, USA) did not increase expression of any of the tested interferon response genes (Appendix A). Collectively, these results demonstrate that the in vitro transcribed artificial pri-miR-31 sequences stimulate the interferon response, which is a likely explanation for the non-specific silencing of HBV gene expression observed (Figure 2A,B).

## 4. Discussion

A functional cure for chronic hepB, defined as loss of detectable HBV surface antigen from the carrier’s circulation, is rare and has taken years of treatment in instances of success [15]. The RNAi pathway offers a means of achieving robust silencing of HBV replication [22,23,24,25,27,42], however translating the impressive silencing achieved at the bench to the clinic remains elusive. This stems from the difficulty of in vivo delivery of nucleic acid sequences to the intended site. Use of chemically synthesised siRNAs has enjoyed much greater success because of the ease of clinical translation of the technology [32,33,34,35,36,37,38]. Although we have previously described artificial pri-miRNA sequences that are capable of effectively silencing HBV replication, safe and efficient delivery of the expression constructs remains a significant hurdle before clinical translation. Recent developments in mRNA vaccine technology offer two important factors for clinical translation of artificial miRNA, namely use of in vitro transcribed RNA as the therapeutic sequence and delivery of the RNA using lipid nanoparticles. Increased interest in the production of in vitro transcribed RNA at the industrial scale has surged with the success of the SARS-CoV-2 mRNA vaccines [45,46]. This has created the ideal opportunity for developing RNAi activators expressed as in vitro transcribed sequences that are delivered using highly efficient lipid nanoparticles. To this end we adapted artificial pri-miR-31 sequences we had developed previously for expression as in vitro transcribed sequences. Inhibition of HBV gene expression by the in vitro transcribed artificial pri-miR-31 sequences was observed, however this appeared to be primarily a result of non-specific gene silencing. A luciferase reporter assay, which eliminated effects from non-specific silencing, demonstrated that the artificial pri-miR-31 sequences were capable of specific silencing, albeit at very low levels. Northern blot hybridisation revealed that in vitro transcribed pri-miR-31/5 was processed to the mature guide at very low levels and that most of the transcripts remained in an unprocessed state, which explains the limited target silencing. Finally, assessment of innate immune stimulation by the artificial pri-miR-31 sequences revealed significant induction of interferon response genes, *IFN-β*, *OAS1* and *MxA*. Although further characterisation would reveal details of the exact mechanism/s by which the interferon response is stimulated our analysis indicates that in their current configuration the artificial pri-miRNA are immunostimulatory. Reducing immunostimulatory effects of the in vitro transcribed sequences and enhancing silencing capacity through improved processing will be key to the clinical translation of this technology.

Endogenous RNAs are extensively modified after transcription and these modifications are important for recognition of these RNAs as self and avoiding immunostimulation [47]. The use of modified nucleotides during in vitro transcription produces transcripts that have reduced immunostimulatory properties [48,49,50]. Synthetic siRNAs produced with chemical modifications are functional, indicating tolerance of the RNAi machinery for RNAs with altered structures [51,52,53]. Furthermore, these modifications often improve the activity and stability of the synthetic RNAi activators. Because to date artificial pri-miRNAs have exclusively been used as expressed RNA sequences, they have not been chemically modified. The effect of modified nucleotides on the functionality of an in vitro transcribed pri-miRNAs is therefore unknown. The secondary structure of pri-miRNAs is important for recognition and processing of these sequences, which may be perturbed by the inclusion of modified nucleotides. A recent study demonstrated that complete substitution of a self-amplifying RNA with specific modified nucleotides was well tolerated and improved expression from this RNA [54]. As the function of self-amplifying RNAs is dependent on secondary structures at their 5′ ends this bodes well for use of modified nucleotides for in vitro transcribed artificial miRNAs. We have shown that artificial pri-miRNAs produced as in vitro transcribed sequences achieve modest inhibition of HBV gene expression. Poor efficacy is likely to be a result of limited processing of the artificial pri-miRNAs by the nuclear microprocessor complex. Enhancing trafficking of the artificial pri-miRNA to the nucleus once it has entered the cell may improve target silencing. Avoiding the need for the first processing step, by designing the RNAi activator as an artificial pre-miRNA, may be a simpler alternative. As pre-miRNAs are processed by Dicer in the cytoplasm there is no need for trafficking the sequence to the nucleus. Achieving significant reduction of circulating HBV antigens represents an important approach to realising a functional cure for chronic hepB. HBsAg seroconversion is an important first step to enable reactivation of inflammatory responses in the liver to control HBV gene expression. Optimising the in vitro transcribed RNA platform for use with artificial pri/pre-miRNAs holds promise to achieve the goal of functional cure.

## Figures and Tables

**Figure 1 microorganisms-13-00604-f001:**
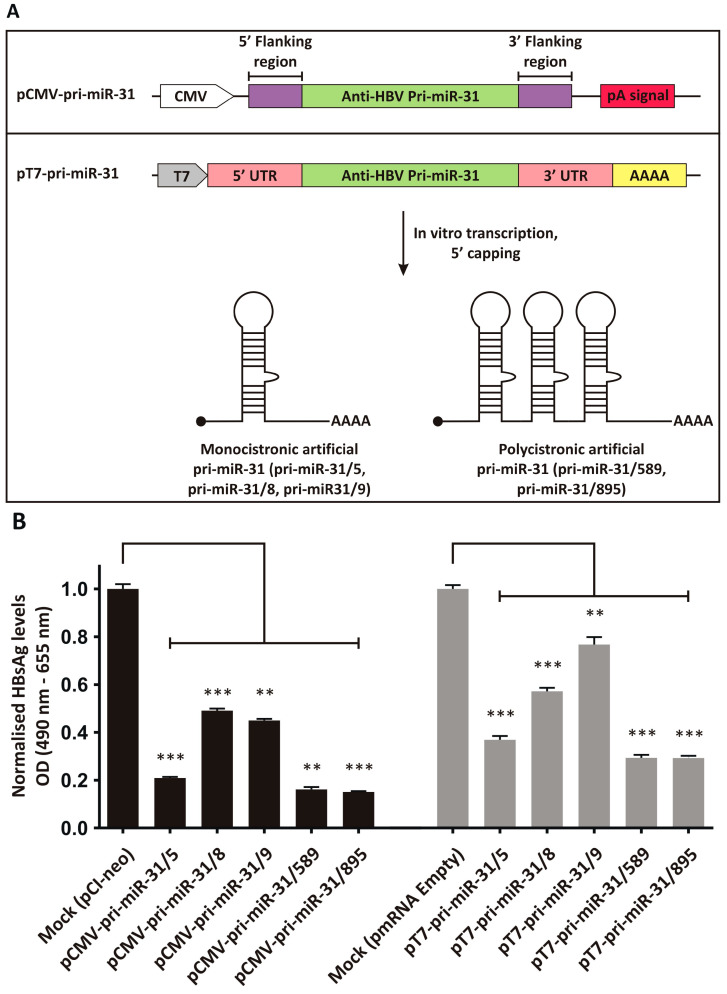
Design and inhibitory capacity of artificial pri-miR-31 expression vectors. (**A**) Upper panel: Original CMV promoter-driven anti-HBV artificial pri-miR-31 vector. Bottom panel: Vector design for in vitro transcription of artificial pri-miR-31 sequences. The vector contains the T7 promoter to enable use of the T7 bacteriophage RNA polymerase. The 3′ end comprises a sequence encoding a 52 nucleotide stretch of adenosines, which makes up the polyA tail. RNAs are capped in a separate enzymatic reaction to add a 7-methylguanosine to the 5′ end. (**B**) HBsAg secretion from HepG2-hNTCP co-transfected with the pri-miR-31 expression vectors (CMV promoter-driven vectors, left and IVT vectors, right) and the HBV target plasmid, pCH-9/3091, were determined by ELISA. The graphs were plotted as the mean ± the standard error of the mean (SEM) (*n* = 3). The statistical significance was calculated using the Student’s paired two-tailed *t*-test (*** indicates *p* < 0.001 and ** indicates *p* < 0.01).

**Figure 2 microorganisms-13-00604-f002:**
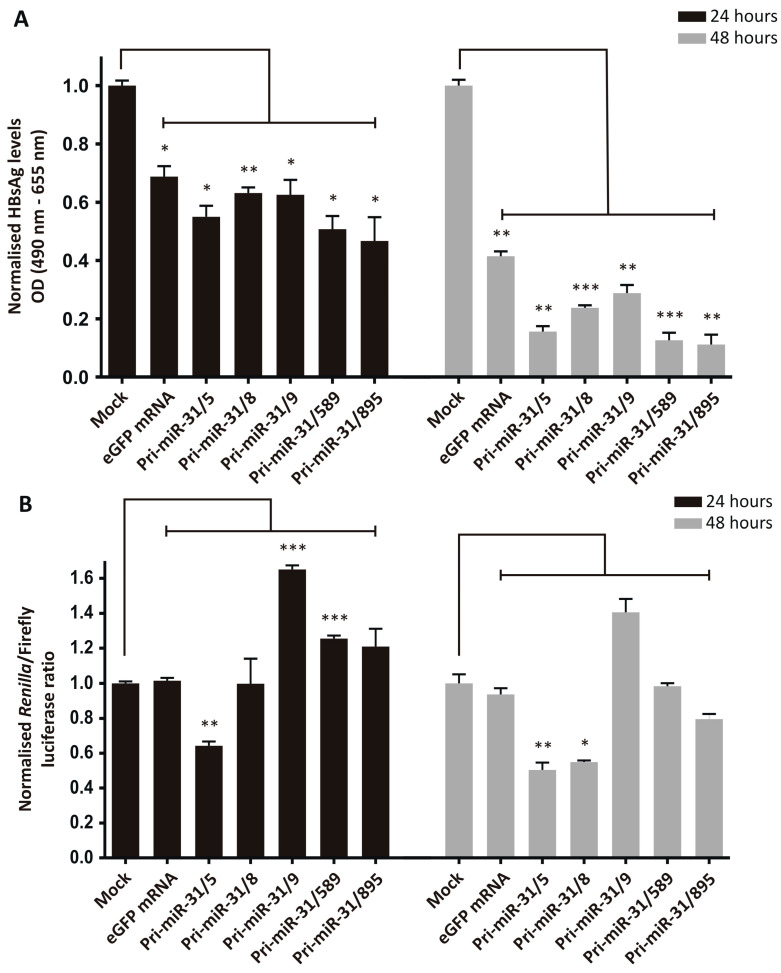
Inhibition of HBV gene expression by in vitro transcribed artificial pri-miR-31 sequences. (**A**) HepG2-hNTCP cells were co-transfected with the HBV target plasmid, pCH-9/3091, and an mCherry reporter plasmid. One day later the cells were transfected with the indicated in vitro transcribed sequences. Secreted HBsAg levels in culture supernatants at 24- and 48-h post-transfection were measured by ELISA and plotted as the mean ± standard error of the mean (SEM) (*n* = 3). The statistical significance was calculated using the Student’s paired two-tailed *t*-test (*** indicates *p* < 0.001; ** indicates *p* < 0.01; * indicates *p* < 0.05. (**B**) HepG2-hNTCP cells were co-transfected with the HBV target plasmid, psiCHECK-HBx, and an mCherry reporter plasmid. One day after the DNA transfection the cells were transfected with the indicated RNA sequences. Twenty-four and forty-eight hours after RNA transfection the cells were harvested and luciferase activity measured in cell lysates. Graphs show the mean ratios of *Renilla* luciferase to Firefly luciferase activities. Error bars represent the SEM. Statistical significance was calculated against the mock control using the Student’s paired two-tailed *t*-test (*** indicates *p* < 0.001; ** indicates *p* < 0.01; * indicates *p* < 0.05).

**Figure 3 microorganisms-13-00604-f003:**
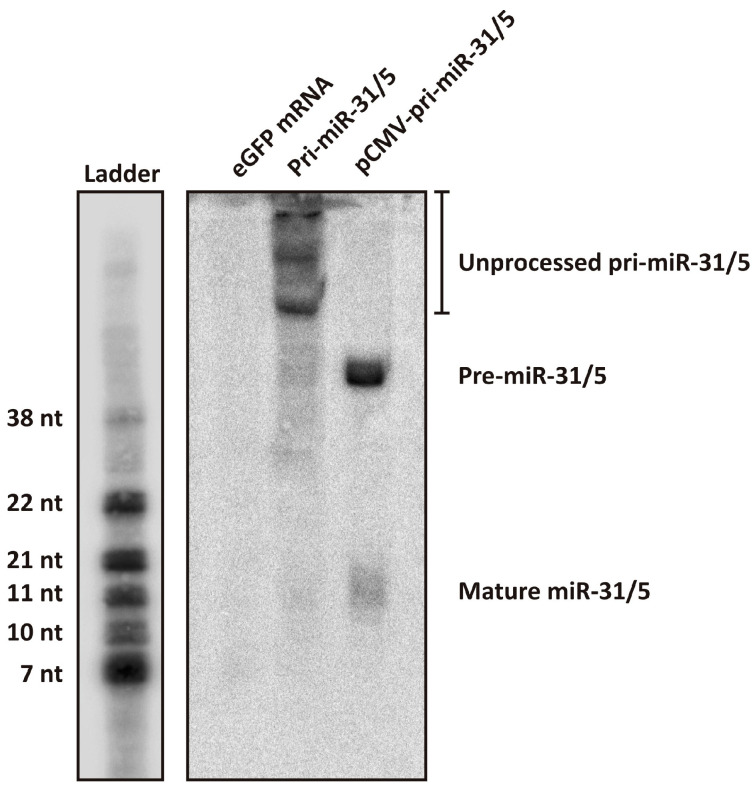
Assessing processing of the artificial pri-miR-31/5 sequence using Northern blot hybridisation. Total RNA was extracted from HEK293T cells that had been transfected with eGFP mRNA, in vitro transcribed pri-miR-31/5, or pCMV-pri-miR-31/5. The samples were separated on a 20% denaturing polyacrylamide gel and Northern blot analysis was performed using a radioactively labelled probe complementary to the expected guide sequence of pri-miR-31/5. The ladder was generated by radioactively labelling several oligonucleotides of different lengths and running them together alongside the RNA samples. The sizes indicated are approximations of the RNA sizes.

**Figure 4 microorganisms-13-00604-f004:**
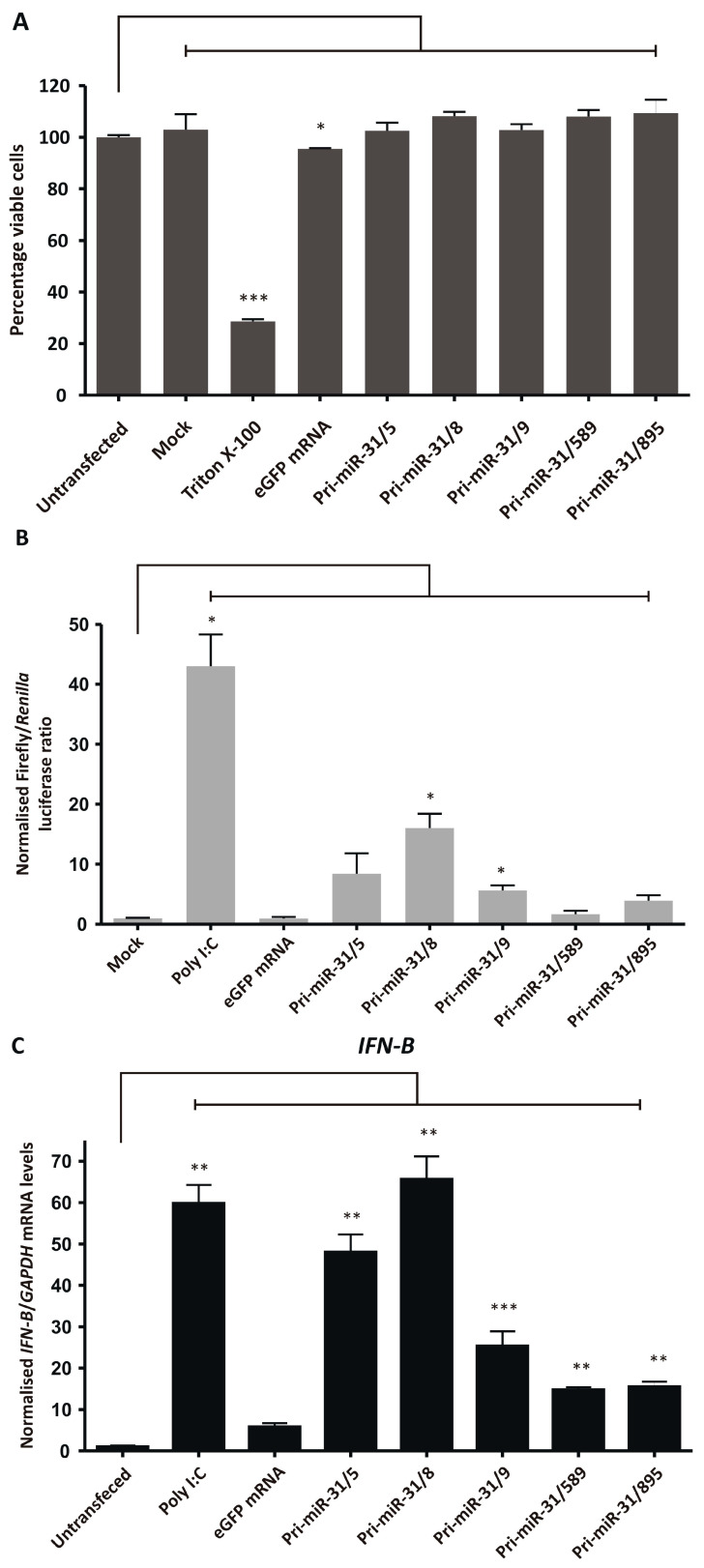
Assessing non-specific effects of in vitro transcribed artificial pri-miR-31 sequences. (**A**) HEK293T cells were transfected with the indicated RNA and their viability assessed by MTT assay 48-h later. The graph was plotted using the mean normalised to the mean of the untransfected cells. Error bars indicate the normalised SEM (*n* = 3). The statistical significance was calculated against the untransfected control using the Student’s paired two-tailed *t*-test (*** indicates *p* < 0.001; * indicates *p* < 0.05). (**B**) HEK293T cells were co-transfected with phRL-CMV, pUCAsc-ISRE×3 and the indicated in vitro transcribed RNA or poly I:C. Two days after transfection, Firefly and *Renilla* luciferase activity was measured using a dual-luciferase assay. The graph shows the mean ratios of Firefly to *Renilla* luciferase activity ± SEM (*n* = 3). Statistical significance was calculated against the mock control using the Student’s paired two-tailed *t*-test (* indicates *p* < 0.05). (**C**–**E**) Expression levels of the interferon stimulated genes *IFN-β* (**C**), *OAS1* (**D**) and *MxA* (**E**) in HEK293T cells transfected with the indicated in vitro transcribed RNA were assessed using qRT-PCR. The graphs represent the relative levels of *IFN-β*, *OAS1* and *MxA* to *GAPDH* mRNA. Each graph is normalised to its respective mock (mean ± SEM, *n* = 3). Statistical significance was calculated against the mock treated cells using the Student’s paired two-tailed *t*-test (*** indicates *p* < 0.001; ** indicates *p* < 0.01; * indicates *p* < 0.05).

## Data Availability

The original data presented in the study are openly available in Figshare at https://doi.org/10.6084/m9.figshare.28524140.

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
