# Peer review of "In Vitro Transcribed Artificial Primary MicroRNA for the Inhibition of Hepatitis B Virus Gene Expression in Cultured Cells†"

_microorganisms, 2025, doi:10.3390/microorganisms13030604_

Round 1

Reviewer 1 Report

Comments and Suggestions for Authors

Chronic hepatitis B management faces challenges due to the limited efficacy of current treatments and obstacles in vaccination efforts. RNA interference has emerged as a promising strategy, enabling the specific silencing of target genes through the introduction of RNA sequences that mimic natural activators of this pathway. This is an important study about developing to target hepatitis B virus (HBV), which would be a great fundament to further use primary microRNAs to block HBV. I suggest accepting this article after minor revisions. Following notes are my comment:

Major concerns:

1.       In many figures (i.e., Fig. 1B, Fig. 2 and Fig. 4), data were analyzed by Student’s t-test, while this is a parametric test, as the data used should be in normal distribution, please ensure all data follow normal distribution in manuscript.

2.       Many figures are confused, readers may not clarify the comparison in figures. For example, in Fig.2, the results of 24h and 48h mixed together, which may cause the confusion. I prefer to divide the results of 24h and 48h to separate figure.

3.       In all figures including the comparison with Student’s t-test, it is better to use line to mark to comparison objects.

Minor concerns:

1.       In title, Mrna should be mRNA.

2.       Extra space in line 23.

3.       Line 189, “Assessment of cellular toxicity” may be the title of the next paragraph, please clarify.

4.       Many errors in references, which should be modified throughout all references.

5.       In the results part, results of Student’s t-test should show with the p value and df.

6.       Replicated “2017” in line 516.

7.       Reference 3, incorrect abbreviation of journal.

8.       Reference 10, incorrect abbreviation of journal.

9.       Reference 23, incorrect abbreviation of journal.

Comments on the Quality of English Language

The English in this paper is well-expressed

Reviewer 2 Report

Comments and Suggestions for Authors

Interesting manuscript as idea, but quite speculative in the Discussion of the results.

Generally, the aim is not well presented and not adequately reported in Discussion. Potential application to the clinical practice is too speculative and the title should introduce this reporting that the study is only an attempt in vitro. Too much unresolved problems remain from in vitro to in vivo and potential application should be discussed. The reader can suspect that it is only an experimental exercise rather than an attempt to find an applicative experimental design.

lines 255-256: "HBsAg is a convenient marker to assess HBV replication because the antigen is produced in high concentrations during replication and can be measured easily by ELISA. " this is not HBeAg correlates with HBV DNA replication rather that HBsAg that is a marker of BHV expression (quite different from replication); HBsAg is less accurate marker of HBV replication than HBeAg. This point should be revised. Moreover, it was not enough what done to investigate immune-interference of the artificial pri-miR-31:

lines 470-474: "Finally, characterisation of innate immune stimulation by the artificial pri-miR-31 sequences revealed significant induction of interferon response genes, IFN-β, 471 OAS1 and MxA. Reducing immunostimulatory effects of the in vitro transcribed sequences and enhancing silencing capacity through improved processing will be key to the 473 translation of this technology towards clinical utility."

Round 2

Reviewer 2 Report

Comments and Suggestions for Authors

The comments were adequately addressed.